# Evaluating a 3D Ultrasound Imaging Resolution of Single Transmitter/Receiver with Coding Mask by Extracting Phase Information

**DOI:** 10.3390/s24051496

**Published:** 2024-02-25

**Authors:** Mohammad Syaryadhi, Eiko Nakazawa, Norio Tagawa, Ming Yang

**Affiliations:** Graduate School of Systems Design, Tokyo Metropolitan University, 6-6 Asahigaoka, Hino 191-0065, Tokyo, Japan; syaryadhi-mohd@ed.tmu.ac.jp (M.S.); nakazawa-eiko@ed.tmu.ac.jp (E.N.); yang@tmu.ac.jp (M.Y.)

**Keywords:** single transducer, coding mask, frequency subband compound, SCM

## Abstract

We are currently investigating the ultrasound imaging of a sensor that consists of a randomized encoding mask attached to a single lead zirconate titanate (PZT) oscillator for a puncture microscope application. The proposed model was conducted using a finite element method (FEM) simulator. To increase the number of measurements required by a single element system that affects its resolution, the transducer was rotated at different angles. The image was constructed by solving a linear equation of the image model resulting in a poor quality. In a previous work, the phase information was extracted from the echo signal to improve the image quality. This study proposes a strategy by integrating the weighted frequency subbands compound and a super-resolution technique to enhance the resolution in range and lateral direction. The image performance with different methods was also evaluated using the experimental data. The results indicate that better image resolution and speckle suppression were obtained by applying the proposed method.

## 1. Introduction

Ultrasound imaging has been widely applied to medical diagnosis due to its superiority via non-ionizing radiation and its relatively low-cost system. At present, it is performed by using an array system, which requires a number of sensors to be properly arranged and applies a general beamforming method. For accurate diagnosis, it should have a high resolution and high signal-to-noise ratio (SNR), and therefore researchers have intensively studied many methods to obtain a better image quality [1,2,3,4,5,6,7,8,9,10].

In the current pathological examination, the tissue diagnosis takes a certain time for its performance. It is started by obtaining the tissue sample of human body by biopsy process which gives a burden to the patient and performs the observation under a standard optical microscope. It can be solved by developing an ultrasonic puncture microscopy which is possible to capture the tissue area of human body directly with easy operation and while being comfortable for the patient. However, such a device should have a simple compact structure system and is not suitable for implementation by a sensor array due to the significant amount of wiring and complex electronic circuit needed to generate the ultrasound image [11,12,13,14].

Recently, structured ultrasound microscopy (SUM) has been proposed by using a single transmitter/receiver circuit [15,16] while placing the coding mask [17]. However, single transducers with a single time-series echo signal required several additional measurements compared to array transducers consisting of numerous elements to construct a 3D image. Therefore, it was integrated by attaching an acoustic mask with irregular abberations to establish spatial coding. Randomizing the thickness of the coding mask leads to a different delay time of the transmission signal traveling inside the mask material and finally to a randomized wave propagation in the target medium. The reflection signals from the scatterers with different delays are also received by the same transducer in a single accumulated echo signal. By rotating the transducer, variations of the measurement have been obtained to increase the spatial coding variation of the region of interest.

Currently, many researchers are developing techniques for improving the ultrasound image resolution based on array transducers [18,19,20,21,22]. The minimum variance beamformer method on the received echo has been proposed to an array system enabling the improvement of image resolution by using adaptive weight [23,24,25]. Instead of that, the imaging methods utilizing a convolution of the transmitted pulse wave and multiple scatterer reflectances to accurately detect scatterer position have also been introduced via Time-Reversal MUltiple SIgnal Classification (TR-MUSIC) [26,27]. It utilizes the principle that echoes received at each element converge on a scatterer when they are time-reversed and retransmitted. The TR-MUSIC detects the received phase of a real scatterer at each element and achieves lateral super-resolution using the MUSIC algorithm. Another technique using a super-resolution FM-chirp correlation method (SCM) to achieve range-MUSIC which uses the noise subspace by subtracting the signal subspace from the total function space of the echo. The fundamental of this method is based on eigenvalue analysis [28].

To improve the image quality, the above methods were adopted and implemented in a single-element system. The minimum variance method with weight-frequency compound was implemented to a single-element system with a single scatterer placed in the region of interest by decomposing the base wide band of echo signal into several narrow subbands by [29]. The method was effective to improve the image resolution in the range direction. However, the resolution in the lateral direction was not significantly improved. Instead of that, the SCM performance showed a better lateral direction resolution. The SCM with a short pulse transmission wave was also implemented adaptively by constructing the image using a linear equation of the image model [30]. Due to the advantages separately, in this study, we proposed a method by integrating both techniques in order to achieve a better image quality. The main contributions of this study can be clearly listed as follows:to develop ultrasound imaging based on single element with coding mask by numerically solving the image model equation,to apply the proposed method by integrating the frequency subband compound and SCM, andto evaluate the image performance by comparing it with other methods by varying the single scatterer position in the region of interest.

## 2. Methods

### 2.1. Image Model

In this study, the measured data in radio frequency (RF) signal form was converted to an in-phase quadrature (IQ) signal for baseband processing. The image model was constructed using the following model
(1)y=Dx,
where *y* is a vector that consists of the time sampling series of the measured IQ echo signal, and the *M*-column vectors of the matrix *D* are the time sampling series of the IQ reflectance signals from each grid in the region of interest (ROI). *D* is assumed to be known a priori. The vector *x* consists of the complex reflectance coefficients corresponding to each grid and is obtained by numerically solving the image model.

### 2.2. Proposed Method: Super-Resolution Weighted Frequency Subbands Compound

Many researchers have studied methods for improving the ultrasound imaging based on sensor array systems. In a previous study, the weighted frequency subbands compound and super-resolution methods have improved the image resolution based on a single element instead of an array system. In this study, both methods have been integrated to be proposed to enhance the image quality as shown in Figure 1.

In this method, the weighted frequency subband compound and the super-resolution method were performed separately. We define the measurement data at *M*-different angles of rotation by $y=[y_{1},y_{2},\ldots ,y_{M}]$ and $D=[D_{1},D_{2},\ldots ,D_{M}]$. The wide base-band of the measured signal was separated into *N* narrow-subbands and *M*-angle of rotations. After applying the IQ detector, the solution-*x* was computed numerically for each subband and angle of rotation, resulting in M×N-images (tensor data *X*). The weighted frequency subband compound was applied to the second-order tensor data *X* by following the minimum-variance distortion-free response (MVDR) method [31]. The details of MVDR can be clearly seen in Appendix A. The method is started by determining the variance-covariance matrix (*R*) from the snapshot vector-$p_{i}\left(t\right)$. In the sensor array, the variance-covariance matrix was obtained by averaging over number of elements. However, in the single-element system, it was achieved by averaging over a number of sensor rotations. The appropriate weight of the subbands compound was adaptively determined based on the variance-covariance matrix and was common for all angles of rotation. The final image of this method for each angle of rotations after feeding back to the previous tensor-*X* is $Y_{MVDR}$ as shown in the figure.

In the present study, SCM was applied to a sensor array system employing a standard beamforming method for image construction. Details of the procedures for applying SCM can be studied in Appendix A. In this study, SCM was applied adaptively to a single element. However, the SCM application based on the single-element system in the previous study was modified by applying the compression process to the matrix-*D* resulting in the equation $D^{H}y=D^{H}Dx$ where H indicates Hermitian transpose. Thus, the SCM profile was calculated based on the modified model and multiplied by the MVDR output, $Y_{MVDR}$. The final image was obtained by applying a simple averaging or coherence factor (CF) [32] to integrate over the number of rotations.

### 2.3. Previous Methods for Image Construction Were Based on a Single Transducer

In this simulation, several methods were performed to evaluate the image quality and compared with the proposed method above. The details of these methods are described below:Method-A: In this method, the image was constructed by numerically solving the linear equation (LE) of the image model, y=Dx, for each angle of rotations [15,16].Method-B: This method uses weighted frequency subband compounds to obtain images for each rotation. The original wide band of the measured signal was decomposed into several narrow subbands with their different center of frequency. The subband decomposition was applied to vector-*y* and matrix-*D* for all rotation angles. The images were constructed by numerically solving a linear equation of the image model with different subbands and angles of rotation. The weight of each subband was calculated and summed up for each angle of rotation using the minimum variance distortion-free response (MVDR) method [29].Method-C: Based on the image model, y=Dx, the SCM profile was extracted from the measured data for each angle of rotation. The final image at different rotation angles was obtained by multiplying the SCM profile with the image-*x* computed by numerically solving the LE [30].Method-D: The image model was modified by applying the *D*-compression to the original image model resulting in $D^{H}y=D^{H}Dx$. The SCM profile was determined based on the modified image model. On the other hand, the solution-*x* of this equation was computed by analytically solving $x={\left(D^{H}D\right)}^{-1}D^{H}y$. The final image of each rotation was achieved by properly multiplying the image-*x* and the SCM profile.

The final image was obtained by simple averaging or CF to integrate the image over the number of rotations.

## 3. Simulation and Experiment Results

### 3.1. Simulation Model

The proposed model was developed using a finite element method (FEM) with OnScale—a simulator for ultrasound propagation and piezoelectric analysis. Figure 2a,b show the simulation model, which consists of a backing layer, a single PZT element, and an encoding mask with different scatterer number positions using water as the medium for propagating the ultrasonic wave. In reality, a physical device is a circular disk transducer that can be implemented to generate 3D images. However, due to the complexity of the computational process, a 2D simulation was performed in this study. Table 1 shows details of the physical parameters used in the simulation.

The important part of the device is a coding mask attached to the surface of the transducer as shown in Figure 2c. It consists of a number of patches with the same width. However, the height of each patch is different with a randomized thickness with a minimum thickness of 0.25λ and a maximum thickness of 1λ that creates a different local wave propagation with different time delays. The coding mask has a function to generate a spatial coding and a focusing transmitted waveform exceeding that of the broken transducer.In this simulation, the transducer was fired by a short pulse with center of frequency (7MHz) (see Figure 3). The snapshot of the FEM simulation in Figure 2c,d clearly shows the effect of the coding mask on the outspreading transmitted wave.

To construct the image based on the model equation, the reflected signal from the scatterers placed in the imaging area represented by the vector-*y* and the grid reflected signal at the ROI of the matrix-*D* were measured. The transducer was rotated at 25 rotation angles to increase the number of the measurements that affect the improvement of the image resolution. The reflected signal of D at the grid position in the ROI was calculated as follows. First, the sound field in the ROI produced by pulse transmission was measured. The ROI for the imaging area was set by 2mm × 2mm. A one-sided acoustic transfer function was calculated from the sound pressure waveforms obtained at each point in the ROI and the transmitted pulse. This was then multiplied by the sound pressure waveform at each point to obtain the reflected signal.

### 3.2. Simulation Result

In this subsection, the simulation results based on the proposed method are presented. After performing the model simulation and measuring data required for solving the image model process, the post-processing and constructed image were performed by MATLAB R2022a. Currently, the image target which is the vector-*x* was obtained by numerically solving Equation (Equation 1) using a least squares method (LSQR), which is available in the MATLAB library, with an iteration residual of 0.1 set in this simulation.

#### 3.2.1. Basic Method: Numerically Solving a Linear Equation of Image Model

In the previous ultrasound wave simulation, the data consisting of the vector *y* and the matrix *D* were collected at different angles of rotation. Figure 4 shows the measured echo signal in RF waveform at different angles of rotation. The RF signal was converted to an analytical signal (IQ) before solving the LE. In this method, the image *x* was constructed by numerically solving the image model for each angle of rotation. The image with a single scatterer at the depth of about 4.5mm was constructed with a different number of angles to evaluate the effect on the image quality as shown in Figure 5a–c. In this simulation, two scenarios were applied to assemble all the images at different rotation angles. The top images were constructed by simple averaging over the number of rotations. While the bottom images were constructed by applying the CF. The images with CF show better speckle suppression compared to the simple averaging method. At present, the image with 25 rotations shows a better image resolution and a good suppression of unwanted signal. However, the image with 15 rotations does not look much different compared to 25 rotations. To confirm this result, the signal intensity distribution crossing the line through the scatterer position in the range and lateral direction of the CF image was also plotted as shown in Figure 5d,e.

#### 3.2.2. Frequency Subbands Compound

In this method, the original baseband echo was decomposed into 35 subbands with a bandwidth of 6MHz. Some decomposed subband signals after applying a Hanning window filter can be seen in Figure 6. It shows that the center frequency of the base band (7MHz) has been changed to a specific frequency for each subband. The decomposition process was performed on the vector *y* and the matrix *D* for all rotation angles. The image model for each subband and rotation angle was solved numerically, resulting in 35×25 images. Figure 7 shows the performance of B-mode images in different subbands at the same number of rotation angle. The image in sub-25 and sub-35 shows poor quality due to its subband being far from the original center of frequency. The image quality for each subband shows a poor resolution and therefore subband merging must be performed to improve it. In a previous study by [29], the narrow bandwidth of subbands showed the better resolution. The narrow bandwidth of the subband shows better resolution in comparison.

In this method, to compound over the number of subbands, we proposed two methods which are simple averaging subbands and weighted frequency subband compound. In the simple averaging method, the image with 35 subbands was averaged for each rotation angle. While in a weighted frequency subband compound, the weight for subbands was calculated and summed up to obtain the image for each rotation angle. The final image was obtained by simple averaging or CF to compound over the number of rotations as in the previous method. Figure 8 shows the B-mode image by simple averaging over the number of subbands and applying weighted frequency subband compounding. The B-mode image by averaging over the number of subbands has better resolution in the range direction compared to the image without subband processing. However, the unwanted signal in the background image is almost the same as before. The image with the CF method is not much different with simple averaging, although it has a better image resolution in the range direction.

The image constructed by a weighted frequency subband averaging shows a better image quality. By simply averaging the compound over the number of rotations, the image resolution in the range direction was improved. It also suppressed more speckle compared to the averaging subbands method. The image quality became better when the CF method was used to average the image at different angles of rotation. The effect of increasing the number of rotation angles also affected the image quality as shown in Figure 9 with simple averaging for rotation compounding. The image with five rotations shows poor resolution compared to the other images. The images with 15 rotations and 25 rotations are not much different to suppress the unwanted signal. In general, the image with the weighted frequency subband combination shows a better image resolution, especially in the range direction, and has a better speckle suppression. Figure 10 shows the effect of the subbands width to the image resolution. The narrow subbands (2 MHz) represent the better image resolution.

#### 3.2.3. Super-Resolution Method (SCM)

In our previous study, the SCM was applied to a sensor array system with a chirp transmitted waveform signal and employed a famous beamforming technique to construct the image. It is effective to improve the image resolution by an eigenvalue analysis approach. In this study, the method was adaptively implemented on a single-element system with a short-pulse transmitted waveform signal, and the image was constructed based on the linear equation of the model.

The SCM was applied to the original image model, y=Dx, for all rotation angles. A variance-covariance matrix was first determined by decomposing the echo signal (*y*) into 35 subbands with a bandwidth of 6MHz. The eigenvalue and eigenvector were extracted from the matrix for eigenvalue analysis, and the eigenvalue was sorted in descending order to detect the signal and noise subspaces. Figure 11 shows the eigenvalue analysis by extracting an eigenvalue profile. From the profile, the eigenvector corresponding to the first eigenvector with a higher value spans a signal subspace. While the eigenvectors corresponding to the next eigenvalues consist of a noise subspace. By assigning each column of the matrix *D* in the image model (y=Dx) as a steering vector, the noise for each grid position was evaluated for 25 rotation angles, resulting in an SCM profile. The final image was obtained by multiplying the profile with the previously numerically solved image.

To avoid the numerical computation of image-*x*, the original image model was modified by applying the compression process to matrix-*D*, resulting in
$$D^{H}y=D^{H}Dx,$$
(2)$$y^{\prime }=Ex,$$
where $y^{\prime }=D^{H}y$ and $E=D^{H}D$. Based on Equation (Equation 2), the SCM profile was computed for all rotation angles as in the previous strategy. Figure 12a,b show the SCM profile at a given rotation angle with different approximations. In the case of a single scatterer, the *C* value was set to 1 to compute the SCM profile. The varying *C* value affected the SCM profile with the appearance of an unwanted signal as shown in Figure 12c. The SCM profile was plotted in the range direction on a line crossing the scatterer position. B-mode images of a single scatterer (*C* = 1) are shown in Figure 13 with simple averaging and CF composite.

#### 3.2.4. Proposed Method

From the results presented before, the image constructed by applying the weighted subband frequency compound has better resolution only in the range direction. While by applying a super-resolution method, the image quality has better improvement in the lateral direction. In the current super-resolution method, the image was obtained by multiplying the image *x* by the computed SCM profile for each angle of rotation. Therefore, we integrated both methods to obtain better image quality as our proposed method. The SCM profile was multiplied by the image obtained by applying the weighted frequency subbands method. Figure 14 shows B-mode images with simple averaging and CF compound, respectively. The B-mode image obtained by varying the bandwidth of subbands can be found in Figure 15 for comparison. To show the qualitative performance of the described methods, Figure 16 shows the amplitude profiles in range and lateral resolution. The cross-sectional profile was obtained by picking up a normalized intensity along the line passing through the target position in both directions of the image with CF compound. The label ‘LE’ represents the image obtained by numerically solving the linear equation of the image model. The method for combining weighted frequency subbands is referred to as ‘Freq. subbands’. The super-resolution method without compression is labeled as ‘SCM’, while the method with compression is labeled as ‘SCMcompr.’. The label ‘Proposed’ indicates the integration of frequency subbands compound and super-resolution methods.

To assess the effectiveness of the methods, we placed a single scatterer at various positions and performed a quantitative analysis and evaluation. A single target with the same properties and size was placed at nine positions in the imaging region. Three parameters were used to evaluate the image characteristics: signal-to-noise ratio (SNR), speckle level, and full width at half-maximum (FWHM). SNR in dB was calculated by dividing the averaged signal by the standard deviation of the noise. Speckle level in dB was determined by averaging the image noise. FWHM was calculated at −3dB of the amplitude profile intersecting the single scatterer of the image with CF compound. Figure 17 shows the SNR, speckle level and FWHM associated with the error bar, respectively. The error bar was calculated by averaging the standard deviation over the number of samples (nine positions).

#### 3.2.5. Evaluate the Methods with Multiple Scatterers

This study also investigated the imaging characteristics by placing multiple targets, specifically five scatterers, within the region of interest (ROI) instead of a single scatterer. The targets were identical in size and properties as the single scatterer. To compute the SCM profile for five scatterers, the C-value in Equation (Equation 17) was set to 5. Figure 18 displays the B-mode images with different methods applied for multiple scatterers. The image obtained by the proposed method with integration between weighted frequency subbands compound and SCM with compression shows better resolution. The evaluation parameters for the case of mutiple scatterers can be found in Figure 19, consisting of the SNR, the speckle level and the FWHM.

### 3.3. Experimental Condition

To confirm the simulation results, we demonstrated the proposed method using experimental data. An array transducer system modeled the single element with a coding mask in this experiment. We used an experimental platform for medical ultrasound applications (RSYS0003, Microsonic Co., Ltd., Tokyo, Japan) to perform the transmit and receive sequence. The sampling frequency was 31.25 MHz, and the maximum number of samples was 1984. The ultrasound transducer used was a linear array probe (T0-1599, Nihon Dempa Koygo Co., Ltd., Tokyo, Japan) with 64 elements (0.2 mm thick and 0.115 mm spacing) transmitting a short pulse signal with a center frequency of 5.21 MHz. Following the ultrasound measurement, signal processing was conducted using MATLAB R2022a.

Figure 20 shows the experimental platform used in this work. We present the experimental results obtained using a soft tissue mimicking phantom (Kyoto Kagaku US-2 multipurpose phantom N-365), with a speed of sound of 1432 m/s (25 °C) and an attenuation of 0.59 dB/cm/MHz. As illustrated in Figure 20, the phantom contains four string wires with the diameter of 0.1 mm. For high computational efficiency, we evaluated the single scatterer located approximately 21 mm from the probe.

### 3.4. Experimental Results

Each element of the array transducer separately transmitted a short pulse waveform which was received by all 64 elements. To model the coding mask, a set of random delay times was applied to shift each of the received echo signals based on the element order. Based on the image model, the vector-*y* was represented by a single echo signal that was obtained by summing up all the received echo signals. We provided 40 sets of random delay times that represent the number of rotations in the real single coding mask transducer. The pure received signal of single element consisting of a short pulse transmitted wave and backscattered wave is shown in Figure 21a. Several echo signals at different sets of random delay times can be seen in Figure 21b–d.

The matrix *D*, with a size of 171 × 51, was defined in the region of the desired scatterer position and for computational purposes. It was started by calculating the delay time between the element and each grid point of the ROI and it was added by a random delay time indicating the time interval within the coding mask of a certain thickness attached to the surface of the single transducer in real-time. The total delay time was used to shift the reference signal which is a pulse transmission waveform. The procedure was repeated for all elements and finally summed up to yield a single signal before being multiplied by a transfer function using frequency domain processing. After obtaining the vector t *y* and the matrix *D* for all sets, the image construction can be performed.

Based on the experimental data, the image was constructed with different methods as in the previous simulation. The residual coefficient for the numerical iteration solution of the image model equation was set to 0.01. A 15-subband with a subband width of 6 MHz was used to compute the SCM and the weighted frequency subband compound. Figure 22 shows the B-modes with simple averaging and CF compound over the 40 angles of rotation.

## 4. Discussion

Generally, ultrasound imaging has been performed by employing a sensor array, which results in more measurement data from each element to receive the backscattered signal. However, in the single transmitter/receiver system, it has a finite measurement limit. Therefore, the device was rotated at several different angles to collect more measurement data. As the device rotates, the coding mask also rotates along it, so that the new interference transmitted signal flows to the medium and collects the other information echo signal from the same scatterer. To obtain more varying new information, the angle of rotation is highly dependent on a randomized structure of the coding mask. Figure 4 shows a view of several measured echo signals at different rotation angles. More measurements affect the unique pixel characteristic of the image as shown in Figure 5.

Based on the results, the original image, which is constructed by solving the linear equation at each angle of rotation without any processing, has been improved. The weighted frequency subband composite shows the best image resolution in the range direction and higher speckle suppression compared to other methods. To achieve better spatial coding, the number of measurements in the single-element system was increased by increasing the number of rotations. Speckle suppression and spatial resolution improved as the number of rotations increased, as depicted in Figure 9 with simple averaging.

The image produced by applying the SCM was effective in terms of range direction resolution. The value of *C* is determined based on the number of scatterers. Higher values result in more noise, as shown in Figure 12c. Furthermore, Figure 13 confirms the effect of the *D* compression, which improved the image resolution in both directions compared to the SCM without compression, resulting in better speckle suppression performance. Therefore, we integrated both methods to enhance image quality. On the other hand, the subband width affects image resolution through the weight frequency compound, which is demonstrated by a narrow band. However, in the proposed method, the subband width has a minimal effect on image resolution, as shown in Figure 15. Therefore, we chose a subband width of 6 MHz for this study. For the final comparison of the methods proposed in this study, Figure 17 presents a quantitative evaluation of the image performance on the single scatterer.

The proposed method demonstrates superior image quality when applied to the five targets, as shown in Figure 18. The integrated method between weighted frequency subband compounds and super-resolution methods shows better image resolution by simple averaging or CF to compound over the number of rotations. The frequency subbands compound and the proposed method show almost the same level in speckle suppression. Figure 19 displays the evaluation based on SNR, level of speckle and FWHM of the multiple scatterers case. The SNR with multiple scatterers shows the same trend as the single scatterer case. In terms of the degree of speckle suppression, the frequency subband shows a slightly higher value compared to the proposed method. In addition, the proposed method shows the highest resolution in both directions as shown by FWHM. It is worth noting that the presence of multiple targets and the resulting reflected signals between scatterers can affect the evaluation parameters, as compared to the previous single target evaluation.

Instead of the simulation, we also demonstrated the proposed methods using the experimental data by modelling a single coding mask transducer with the available array transducer system. The constructed B-mode images in Figure 22 confirm the qualitative evaluation of the proposed method, which is effective in improving the image resolution. In the future, the quantitative evaluation of the proposed method using the experimental data obtained from the real single coding mask transducer system can also be achieved as the simulation results show. Furthermore, the current experimental results provide evidence that the proposed method can be implemented in the real single-coding mask system in the next stage of our research.

## 5. Conclusions

The study constructed an ultrasonic image of an encoding mask using a single oscillator. The imaging resolution was evaluated using several methods, including weighted frequency subbands, the super-resolution technique, and our proposed method by integrating both. The final image was obtained by simple averaging or using the coherence factor across 25 different angles of rotation. The highest resolution image was achieved through weighted frequency subband in the range direction. Meanwhile, a super-resolution method demonstrated the highest lateral resolution. Our proposed method combines weighted subband frequency compounding and super-resolution techniques, resulting in the highest quality image. The quality of image was assessed by calculating FWHM, SNR, and speckle level. In the future, experiments will be carried out with a real single-coding mask transducer system to confirm the current results with a qualitative and quantitative evaluation using the proposed method. 

## Figures and Tables

**Figure 1 sensors-24-01496-f001:**
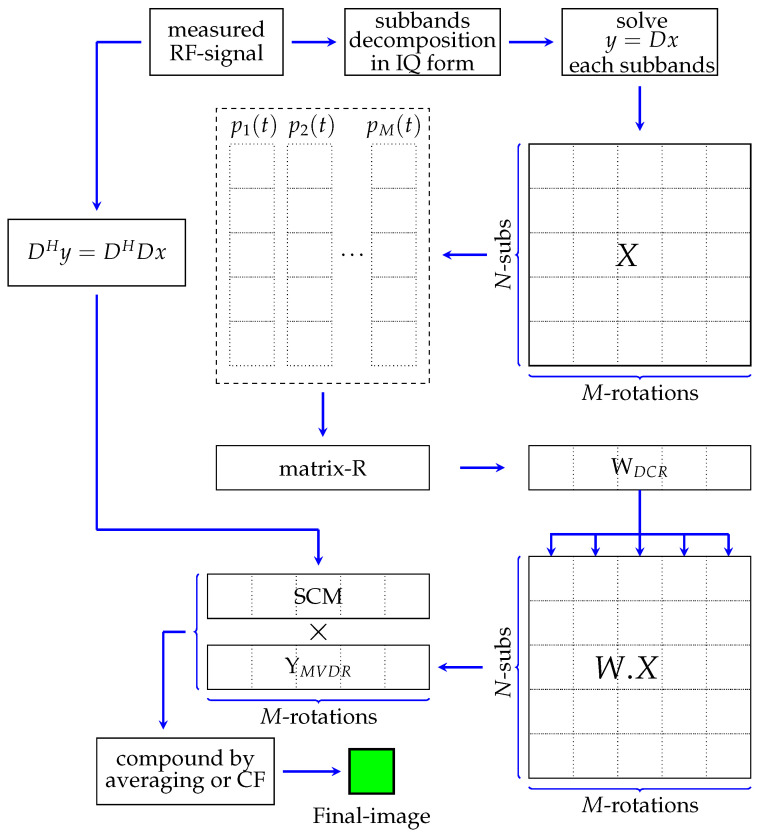
Proposed method by integrating the weighted frequency subband compound and super-resolution method (SCM).

**Figure 2 sensors-24-01496-f002:**
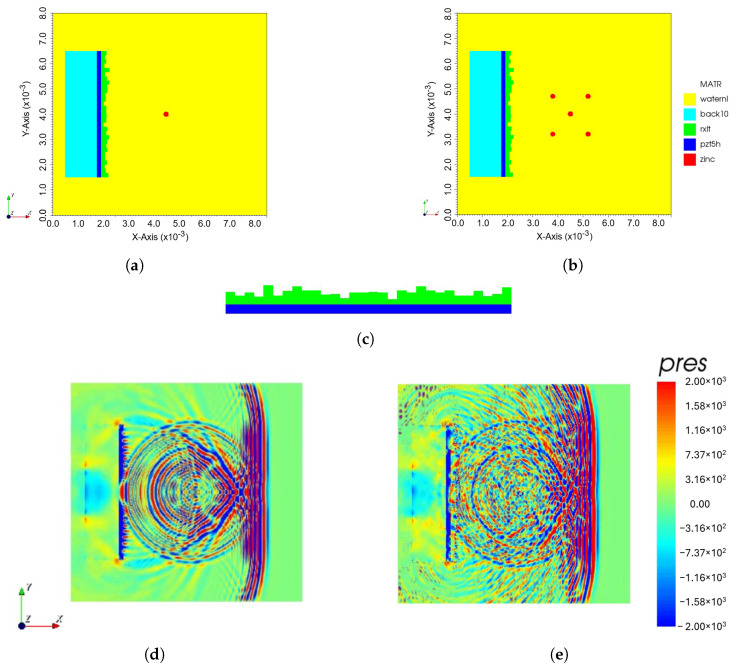
Proposed simulation model with (**a**) single scatterer, (**b**) multiple scatterers, (**c**) side view of coding mask attached to top of transducer, screenshot of FEM simulation (**d**) without coding mask and (**e**) with coding mask of single scatterer.

**Figure 3 sensors-24-01496-f003:**
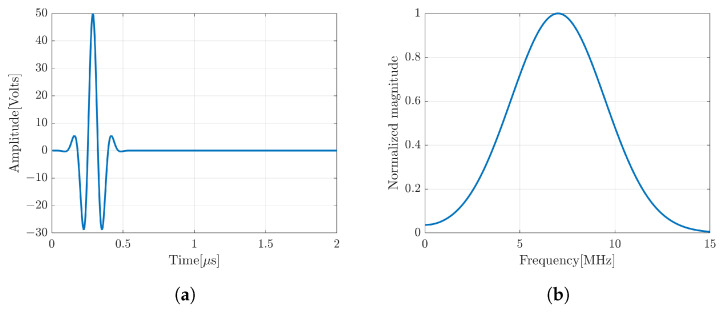
(**a**) Short pulse transmission waveform and (**b**) its center of frequency at 7 MHz.

**Figure 4 sensors-24-01496-f004:**
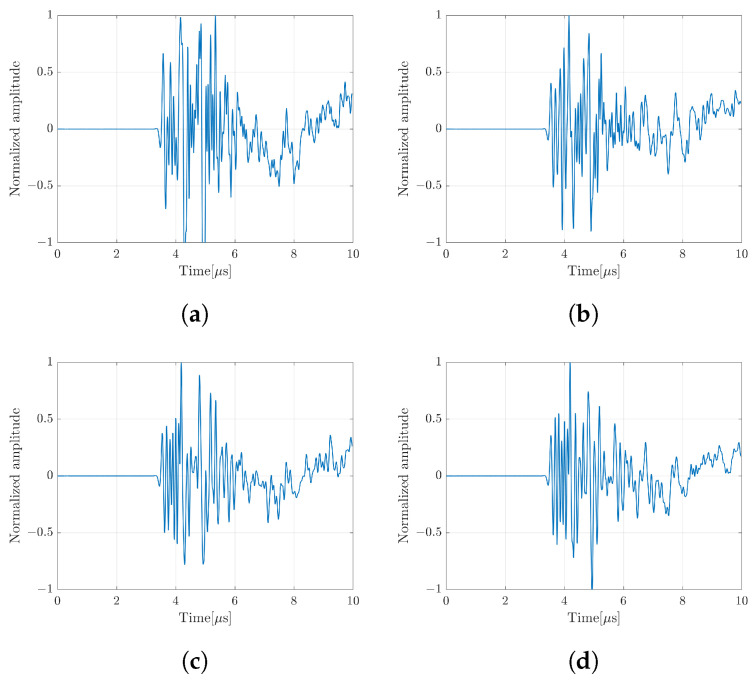
Several echo signals at different angles of rotation; (**a**) angle#1; (**b**) angle#2; (**c**) angle#3; and (**d**) angle#4.

**Figure 5 sensors-24-01496-f005:**
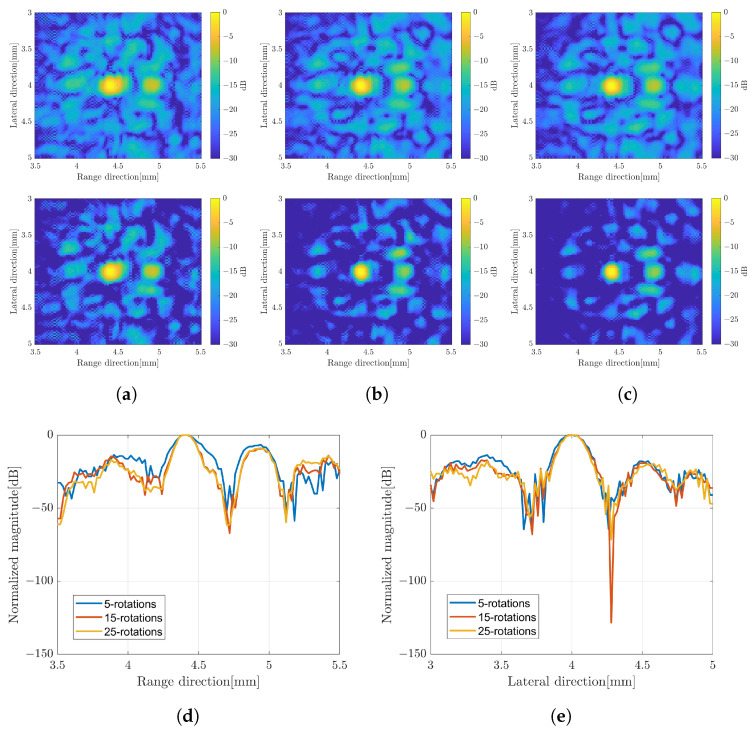
B-mode images constructed by numerically solving the linear equation (LE) with different numbers of rotations, simple averaging (top) and CF composite (bottom); (**a**) five-rotations, (**b**) 15-rotations, and (**c**) 25-rotations. Amplitude profile crossing the scatterer position in (**d**) range and (**e**) lateral direction.

**Figure 6 sensors-24-01496-f006:**
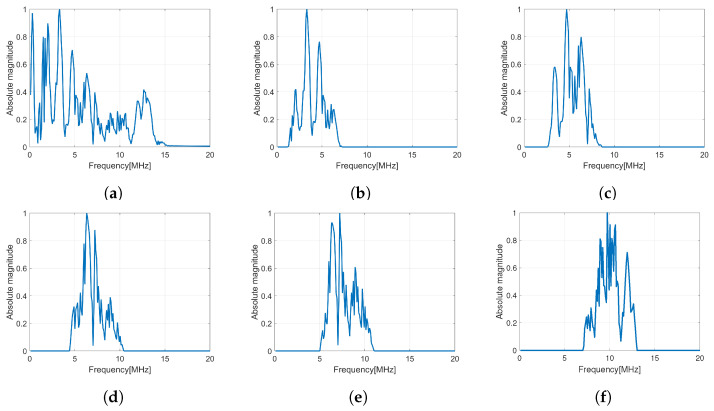
Subband decomposition of a wideband base signal (**a**) into several narrow subband signals in (**b**–**f**).

**Figure 7 sensors-24-01496-f007:**
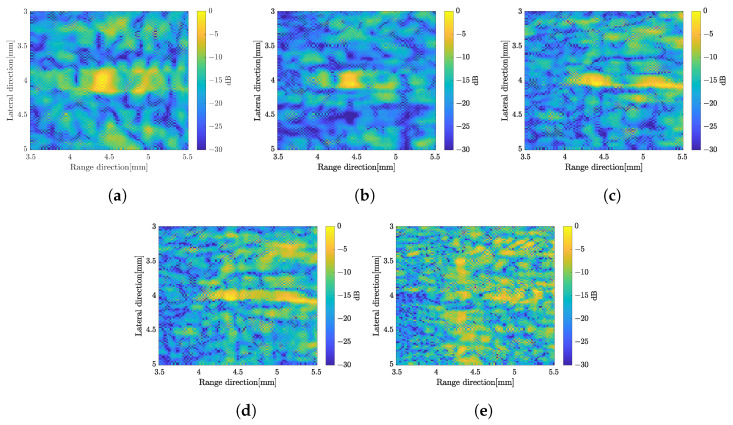
B-modes on different subbands with the same rotation angle; (**a**) sub-1, (**b**) sub-8, (**c**) sub-17, (**d**) sub-25, (**e**) sub-35.

**Figure 8 sensors-24-01496-f008:**
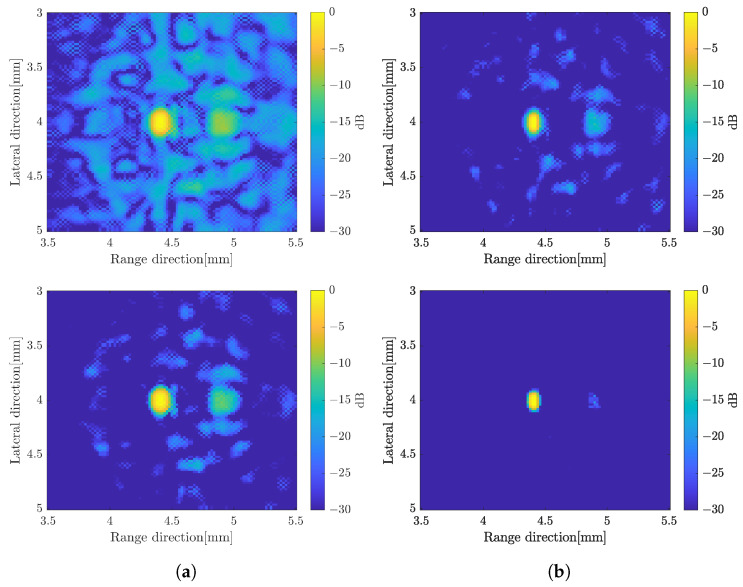
B-mode images by applying the frequency subbands compound, simple averaging (**up**) and CF compound (**down**); (**a**) averaged subbands and (**b**) weighted frequency subbands compound.

**Figure 9 sensors-24-01496-f009:**

B-mode images by weighted frequency subband method with simple averaging for (**a**) five-rotations, (**b**) 15-rotations, and (**c**) 25-rotations.

**Figure 10 sensors-24-01496-f010:**
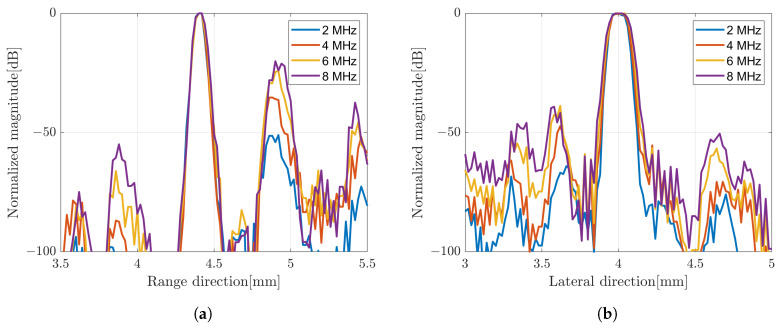
Amplitude profile of B-mode images with different bandwidth of subband; (**a**) range and (**b**) lateral direction.

**Figure 11 sensors-24-01496-f011:**
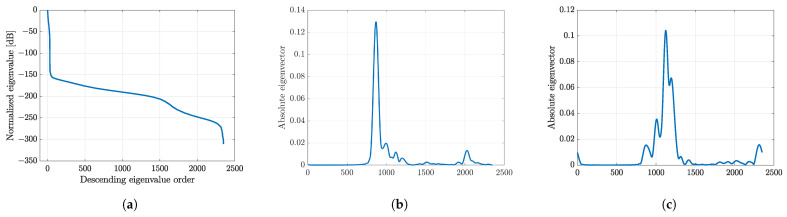
Eigenvalue analysis; (**a**) eigenvalue distribution in descending order with (**b**) the first eigenvector and (**c**) the second eigenvector corresponding to its eigenvalue, respectively.

**Figure 12 sensors-24-01496-f012:**
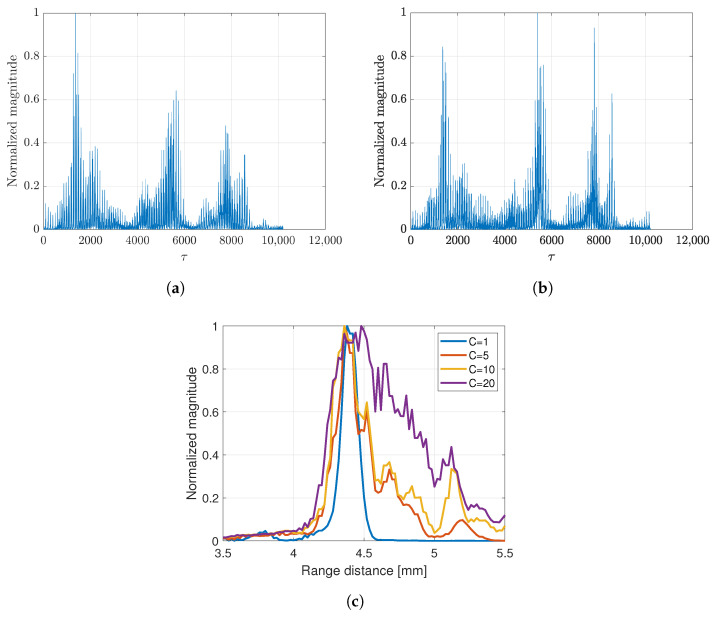
SCM profile (**a**) without compression and (**b**) with *D*-compression. Effect of different *C*-values on (**c**) SCM profile in range direction.

**Figure 13 sensors-24-01496-f013:**
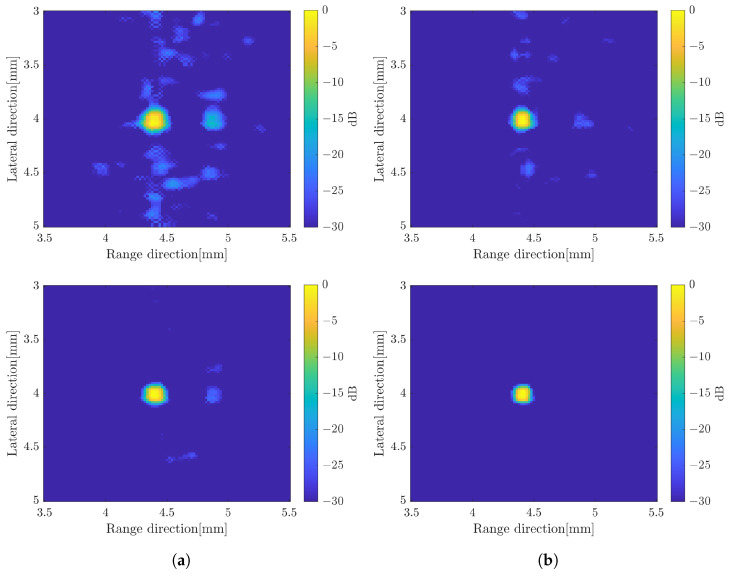
B-mode images by applying the super-resolution method, simple averaging (**top**) and CF compound (**bottom**), (**a**) without compression; (**b**) with the compression process.

**Figure 14 sensors-24-01496-f014:**
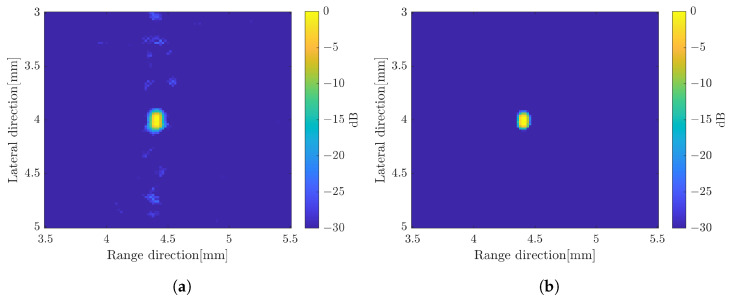
B-mode images by integrating weighted frequency subbands and SCM with (**a**) simple averaging; (**b**) CF compound.

**Figure 15 sensors-24-01496-f015:**
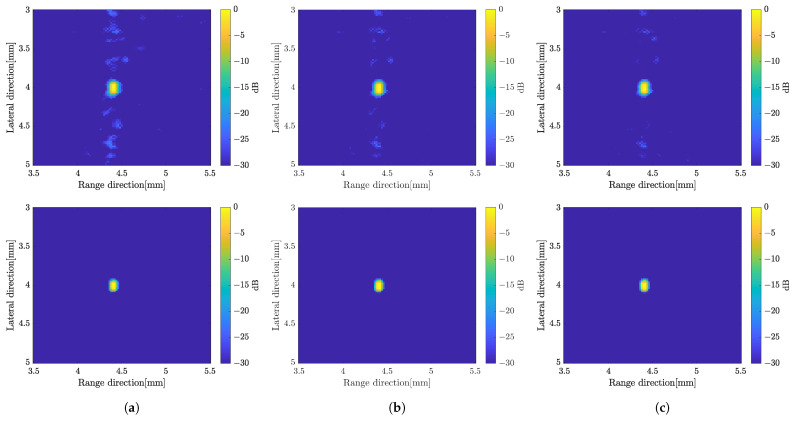
B-mode images constructed by the proposed method with the different width of the subbands; (**a**) 4 MHz; (**b**) 6 MHz; (**c**) 8 MHz with simple averaging (**top**) and CF (**bottom**).

**Figure 16 sensors-24-01496-f016:**
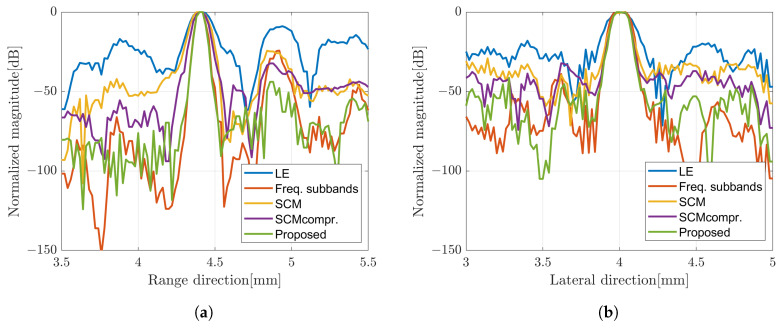
Amplitude profile of B-mode image of CF compound in (**a**) range and (**b**) lateral direction.

**Figure 17 sensors-24-01496-f017:**
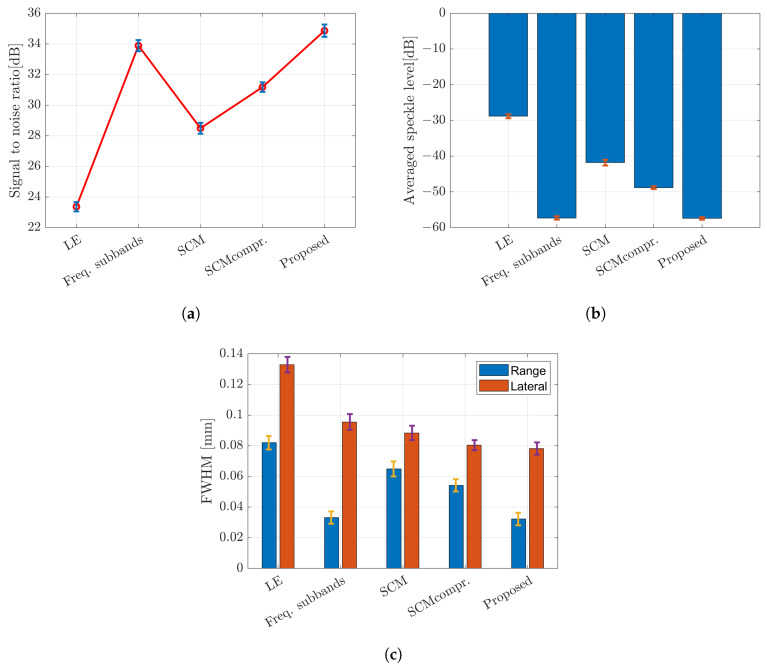
Evaluation parameters with the error bar; (**a**) signal-to-noise ratio (SNR); (**b**) speckle level; and (**c**) FWHM of a single scatterer case placed at nine different positions.

**Figure 18 sensors-24-01496-f018:**
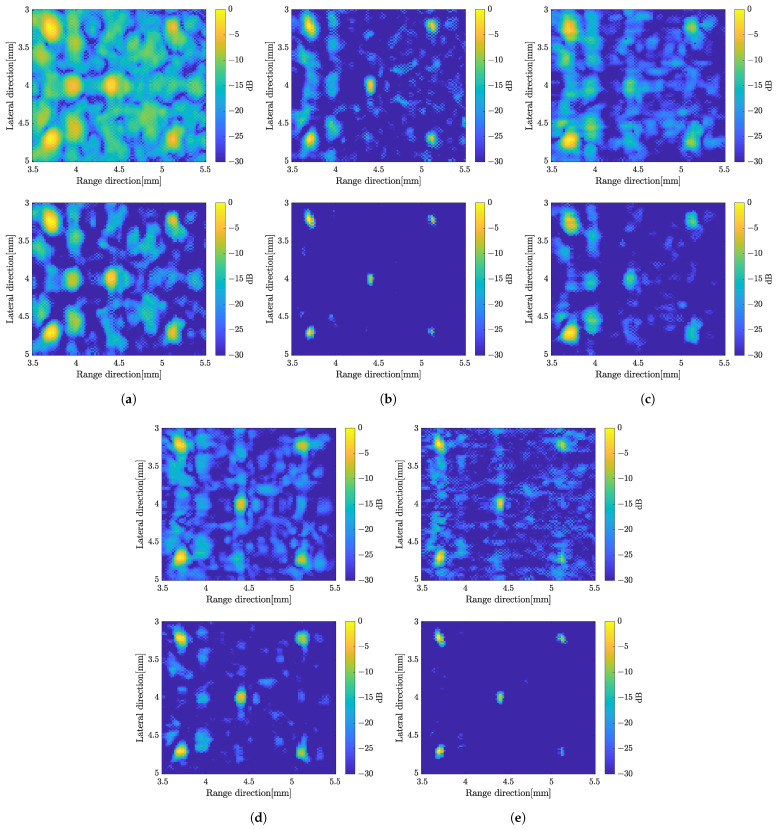
B-mode images with multiple scatterers with simple averaging (**top**) and CF compund (**bottom**), (**a**) solving the original linear equation model, (**b**) weighted frequency subband compound; (**c**) applying the super-resolution method; (**d**) super-resolution method with compression; (**e**) proposed method.

**Figure 19 sensors-24-01496-f019:**
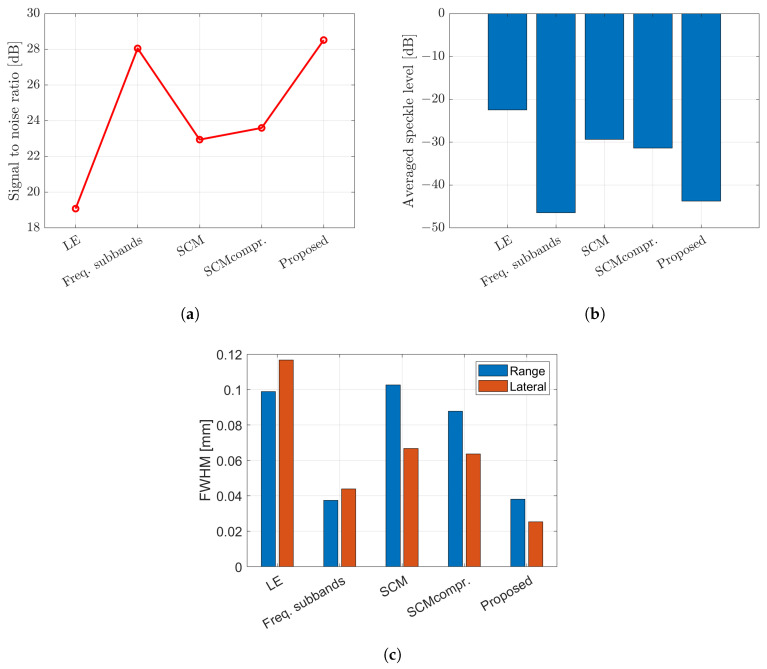
Evaluation parameters for multiple scatterer case; (**a**) signal-to-noise ratio (SNR); (**b**) speckle level; and (**c**) FWHM.

**Figure 20 sensors-24-01496-f020:**
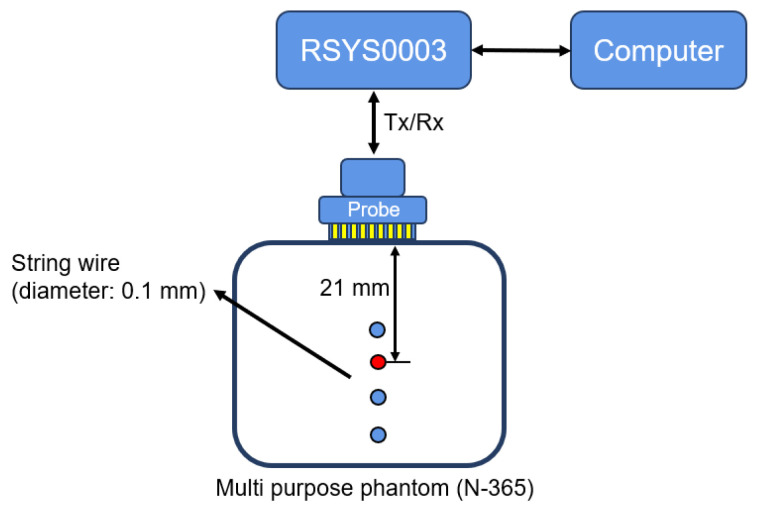
Experimental setting for the soft tissue-mimicking phantom.

**Figure 21 sensors-24-01496-f021:**
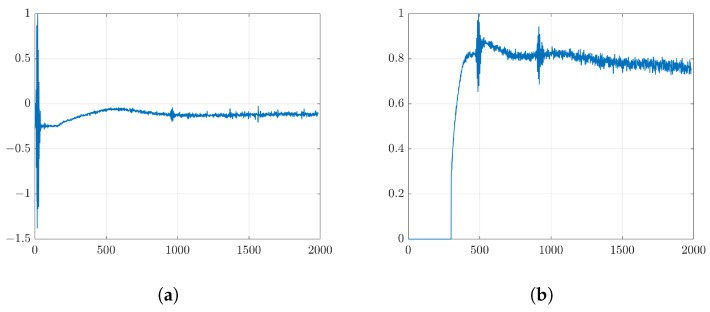
Echo signals of the experiment; (**a**) pure received echo signal of a single element consisting of a short pulse waveform; sum of received echo signals for all elements with (**b**) set#1; (**c**) set#2 and (**d**) set#3.

**Figure 22 sensors-24-01496-f022:**
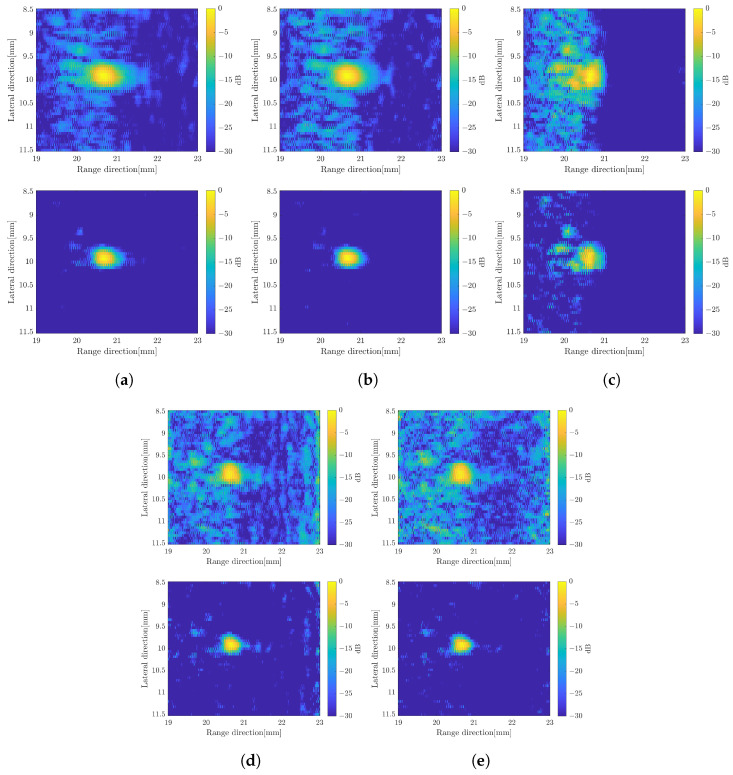
B-mode images using experimental data with simple averaging (**top**) and CF compund (**bottom**), (**a**) numerical linear equation solution method, (**b**) weighted frequency subband compound; (**c**) super-resolution method; (**d**) super-resolution method with compression; (**e**) proposed method.

**Table 1 sensors-24-01496-t001:** Physical parameters used in the simulation.

Parameter	Value
Short transmission pulse voltage	50 Volts
Device length	5 mm
Center of frequency	7 MHz
Backing thickness	1.25 mm
PZT transducer:	
- thickness	0.165 mm
- density	7500 kg/m^3^
- dielectric constant	1700
Coding mask:	
- material	Plastic
- density	1060 kg/m^3^
- bulk velocity	2340 m/s
- number of patches	30
- randomized thickness	0.083–0.335 mm/(0.25λ–1.00λ)
Scatterer radius	0.1 mm
Distance from the scatterer to the surface of the transducer	2.5 mm
Region of interest (ROI) size	2 mm × 2 mm

## Data Availability

The raw data supporting the conclusions of this article will be made available by the authors on request.

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
