# Peer review of "Evaluating a 3D Ultrasound Imaging Resolution of Single Transmitter/Receiver with Coding Mask by Extracting Phase Information"

_sensors, 2024, doi:10.3390/s24051496_

Round 1

Reviewer 1 Report

Comments and Suggestions for Authors

This paper aims to improve the ultrasonic imaging quality of a rotated single element transducer with a coding mask simply by integrating the frequency sub-band compound and super resolution method. The novelty of this work is not well-demonstrated, and the manuscript still need some major revisions.

1. The raw time-domain signals captured at different rotation angles should be presented.

2. In the 2D simulation, how the coding mask rotation is implemented? A 3D model of the coding mask should be presented for ease of understanding.

3. in Figures 4 and 8, a maximum of 25 rotation angles are considered. If these rotation angles evenly are distributed in a 360-degree range? In the cases of 5- or 15-rotations, which 5 or 15 rotation angles are selected?

4. Why a 6 MHz bandwidth is chosen to decompose the echo signal into many sub-bands? How to choose this parameter to achieve an optimal image quality?

5. Experiment results are quite important to verify the simulation results, and should not be omitted.

6. In the simulation, the irregular surface of the coding mask is perfectly attached onto the solid plastic material under test. Practical solid materials do not have such surfaces conformal to the coding mask. How to realize stable acoustic coupling between the two domains?

7. As shown in Figure 14, improvements in SNR, speckle level, FWHM of B-mode images are not significant when using the proposed method that integrates the frequency sub-band compound and SCM.

Author Response

Dear reviewer,

Thank you for contributing your precious time to review our article to be better. The following is the response according to the comments.

  1. We have added a few times domain echoes signal at different angle rotations presented in Figure 4 in the revised manuscript.
  2. In the real implementation of the system, the device is a circular disk of PZT oscillator which possible to generate a 3D image. However, in this simulation, the 2D simulation was performed due to a complex computation. In 2D simulation, the transducer rotation was implemented by different side views of the coding mask to the same scatterer position at different angles of rotation. More measurement data or information is obtained by changing the transducer with coding mask in the rotating direction.
  3. In this single transducer system, each measurement shows spatial information. As the transducer rotates, the interference pattern of the coding mask also rotates along with it and the new information is collected at each measurement at a different angle of rotation. There is no specific rule to determine the angle of rotation adopted in this study. In cases of 5- or 15-rotation angles, the 15-rotation angles are selected to obtain a better image resolution due to more measurement data or more information variation collected. The additional explanation about it has been added to the revised manuscript in the discussion section.
  4. In reference [28], it proposed the weighted frequency subbands compound with different bandwidth. The narrow bandwidth showed a better image resolution compared to the wider bandwidth by the frequency subband compound method. We added the amplitude profile with different bandwidth of subbands (Figure 10) using the frequency subband compound in the revised manuscript. However, in this study, the effect subbands width was not much showing a difference. We have added the B-mode image by the proposed method with different width of subbands in Figure 15 of the revised manuscript.
  5. The next step is to perform the experiment to implement the proposed method using the simulation. Currently, we are setting up the experiment of a single transducer with coding mask and consume time. Our third party, which is specialized in ultrasound technology devices, is fabricating a single circular disk transducer with the coding mask for our next experiment. The diameter of the transducer is 10 mm, and the sound speed of the coding mask is 2500 m/s. However, we are not able to provide the experimental results to confirm the simulation at present. We plan to submit the result in the next report.
  6. We refer to the [16] that successively fabricated the single element with coding mask. The coding mask was made of an 11 mm thin circular plastic layer (Perspex) which has good properties such as high sound speed and low acoustic absorption. The thin water film was sandwiched between the piezo material and the coding mask material to achieve the optimal coupling. In addition, we also received information from the third party that provided our transducer, which specializes in manufacturing the ultrasonic device. They employed a type of adhesive to integrate the coding mask on the top of the PZT element. The adhesive is a silyl group consisting of a polymer.
  7. We agree with your statements. However, the both methods have a different advantages and therefore by integrating, a high image resolution in range and lateral direction can be achieved simultaneously. In the future, we will enhance the image resolution of a single transducer system by applying a proper method. We have added the explanation about it in the introduction section of the revised manuscript.

Finally, we hope that this response has answered the review comment requirements. Along with this response, we have also attached the highlighted manuscript with the changes regarding to the review response.

Reviewer 2 Report

Comments and Suggestions for Authors

The paper proposes a method to improve ultrasound image resolution by integrating the existing methods of weighted frequency sub-bands compound and a super-resolution method named SCM. Furthermore, to increase the number of measurements with a single element transducer, the transducer was rotated at different angles. I have some major concerns regarding the contribution of this paper:

-          The method of rotating the transducer can improve the resolution based on the analysis performed in [1]. This type of analysis is not seen in this paper neither any different analysis and reasoning on the effect of transducer rotation on the resultant image quality.

[1] Hakakzadeh, Soheil, et al. "Multi-angle data acquisition to compensate transducer finite size in photoacoustic tomography." Photoacoustics 27 (2022): 100373.

-          The basic idea of the paper is to improve resolution by integrating two existing methods. So what is the novelty of the paper? There seems no in depth reasoning on why integrating these two methods could improve the resolution more than each individual method.

-          The resolution is of an imaging system is a physical property of that system which is determined by the transducer's size, transducer's spacing, sampling rate, frequency and …. . When a data is captured no post-processing method could improve resolution. The authors should physically analyze the reasons of improving resolution with their proposed method.

Comments on the Quality of English Language

some typographical mistakes are seen in the manuscript. The authors should re-read the paper and edit these mistakes.

Author Response

Dear reviewer,

Thank you for contributing your precious time to review our article to be better. The following is the response according to the comments.

  1. In this single transducer system, each measurement gives spatial information. As the transducer rotates, the interference pattern of the coding mask also rotates along with it and the new information is collected at each measurement at a different angle of rotation. As the number of rotations increases, more information about the imaged region is collected. Eventually, each pixel of the constructed image will have a more unique characteristic as the rotation angle measurement increases.
  2. In a previous study, both methods were implemented in array transducer system with a beamforming technique. In this study, the methods were applied in single oscillator system. The weighted frequency subbands compound (Method-B) improved the image resolution in range direction significantly compared to the SCM (Method-D). However, SCM improved the lateral resolution better than frequency subbands compound. Therefore, we integrated both methods to improve the resolution in both directions as shown in Figure 16c.
  3. The image resolution by solving the linear equation shows the poor quality. To address the low image resolution, the weighted frequency subband compound and SCM was introduced. The better image resolution in range direction was shown by weighted frequency subband compound. By this method, the lateral resolution was not effective. By SCM, the resolution was improved in both directions, especially in lateral direction as desired. Therefore, we integrated both methods to obtain improvement in both directions.

Finally, we hope that this response has answered the review comment requirements. Along with this response, we have also attached the highlighted manuscript with the changes regarding to the review response.

Reviewer 3 Report

Comments and Suggestions for Authors

The authors are currently exploring the ultrasound imaging capabilities of a sensor comprising a randomized encoding mask attached to a single lead zirconate titanate (PZT) oscillator for use in a puncture microscope. The proposed model was executed using a finite element method (FEM) simulator. To enhance the number of measurements required by a single-element system, the transducer underwent rotation at various angles. Image construction involved solving a linear equation of the image model. In a preceding study, image resolution was enhanced through the extraction of phase information. In this investigation, they introduce a strategy that integrates weighted frequency subbands compound and a super-resolution method (SCM) to further improve both range and lateral resolution. Various methods were applied to enhance image quality, and the results indicate superior image resolution and speckle suppression with the proposed approach. This is a nice draft. Some comments are as follows:

1.      The font size in Figure 2 is considerably small, making it challenging for readers. We recommend increasing the font size for better readability.

2.      Error bars are not present in Figure 14a. To enhance the data presentation, please include error bars to represent the uncertainty associated with the data points.

3.      The discussion section lacks experimental verification, and it would be beneficial to include more comparisons with existing data from the literature. This additional context will strengthen the discussion and provide a more comprehensive understanding of the research findings.

Comments on the Quality of English Language

Minor editing of English language required

Author Response

Dear reviewer,

Thank you for contributing your precious time to review our article to be better. The following is the response according to the comments.

  1. The font size in Figure 2 has been revised. The revised Figure 2 can be found in the manuscript.
  2. The error bars have been added in Figure 14a. The revised figure can be seen in Figure 16 of the manuscript.
  3. The next step is to perform the experiment to implement the proposed method using the simulation. Currently, we are setting up the experiment of a single transducer with coding mask and consume time. Our third party, which is specialized in ultrasound technology devices, is fabricating a single circular disk transducer with the coding mask for our next experiment. The diameter of the transducer is 10 mm, and the sound speed of the coding mask is 2500 m/s. However, we are not able to provide the experimental results to confirm the simulation at present. We plan to submit the result in the next report.

Finally, we hope that this response has answered the review comment requirements. Along with this response, we have also attached the highlighted manuscript with the changes regarding to the review response.

Round 2

Reviewer 1 Report

Comments and Suggestions for Authors

1. What do "A, B, C, D, proposed" mean in Figure 14? Descriptions of them should be given.

2. Figure 14 only analyzes the SNR, speckle level, FWHM of the case with one single scatter, which is simple and not convincing enough. Results for the case with multiple scatters shown in Figure 15 should also be presented.

3.  It is regrettable that no experiment is added to the revised manuscript.

Author Response

Dear reviewer,

Thank you for contributing your precious time to review our article to be better. The following is the response according to the comments.

  1. Method-A is the B-mode image generated by numerically solving the linear equation (LE) of the image model. Method-B generates the B-mode image through the weighted frequency subbands compound. Method-C constructed the B-mode image by SCM. Method-D constructed the B-mode image by SCM with compression. Our proposed method, Method-proposed, generated a B-mode image constructed by integrating the weighted frequency subband compound and SCM with compression. To avoid confusion with these labels (A,B,C,D) in Figure 14, we have changed the labels and added them in the revised manuscript in Figure 16 and Figure 17 on page 13. An explanation of these labels can be found on page-12 of the revised manuscript.
  2. Thank you for your suggestion. We have evaluated the case of multiple scatterers with SNR, speckle level and FWHM with B-mode image with CF compound over the number of rotations and added it as Figure 18 in the revised manuscript on page 15.
  3. We have provided the experimental data as suggested before. Our single coding mask system is still in the calibration process in a few months and unable to conduct the experiment at this time. Therefore, the available array transducer was modelled as a single coding mask to verify the proposed method in the simulation results. We have added subsections 3.3 and 3.4 to present the experimental results on page-14 to page-16 in the revised manuscript. 

Finally, we hope that this response has answered the review comment requirements. Along with this response, we have also attached the highlighted manuscript with the changes regarding to the review response.

Reviewer 2 Report

Comments and Suggestions for Authors

The paper has been modified and developed, presenting the novelty and contribution of the work more clearly. Therefore, the paper is acceptable in my view point with the following minor revisions:

1. English Grammar check is required.

2. The results presented in Fig. 18 should by analysed more quantitatively to clearly demonstrate the superior features of the proposed method compared to others. 

Comments on the Quality of English Language

a lot of grammatical and typographical errors are seen through out the paper. for example:

the quantitative evaluation of the image performance of the single element system with for all methods is shown in Figure 17.

Author Response

Dear reviewer,

Thank you for contributing your precious time to review our article to be better. The following is the response according to the comments.

  1. English grammar is checked and proofread. The errors on the manuscript (Page-15) have been revised.
  2. Thank you for your suggestion. We have evaluated the case of multiple scatterers with SNR, speckle level and FWHM with B-mode image with CF compound over the number of rotations and added it as Figure 19 in the revised manuscript on page 15. The proposed method shows better results compared to others.

Finally, we hope that this response has answered the review comment requirements. Along with this response, we have also attached the highlighted manuscript with the changes regarding to the review response.